# What proportion of people have a follow-up biopsy in randomized trials of treatments for non-alcoholic steatohepatitis?: A systematic review and meta-analysis

Dimitrios A. Koutoukidis[1,2]*, Elizabeth Morris[1], John A. Henry[1], Yusra Shammoon[1], Matthew Zimmerman[1], Moscho Michalopoulou[1,2], Susan A. Jebb[1,2], Paul Aveyard[1,2]

1 Nuffield Department of Primary Care Health Sciences, University of Oxford, Oxford, United Kingdom,
2 NIHR Oxford Biomedical Research Centre, Oxford University Hospitals, NHS Foundation Trust, Oxford, United Kingdom

* dimitrios.koutoukidis@phc.ox.ac.uk

## Abstract

### Background and aim

Trials of treatments for non-alcoholic steatohepatitis require endpoint assessment with liver biopsies. Previous large-scale trials have calculated their sample size expecting high retention but on average did not achieve this. We aimed to quantify the proportion of participants with a valid follow-up biopsy.

### Methods

We conducted a systematic review of MEDLINE and Embase until May 2020 and included randomized clinical trials of any intervention in non-alcoholic steatohepatitis with at least 1-year follow-up. We were guided by Cochrane methods to run a meta-analysis with generalized linear mixed models with random effects.

### Results

Forty-one trials (n = 6,695) were included. The proportion of participants with a valid follow-up biopsy was 82% (95%CI: 78%-86%, $I^2$ = 92%). There was no evidence of a difference by location, trial length, or by allocated treatment group. Reasons for missing follow-up biopsies were, in ranked order, related to participants (95 per 1,000 participants (95%CI: 69–129, $I^2$ = 92%), medical factors, protocol, trial conduct, and other/unclear. Biopsy-related serious adverse events occurred in 16 per 1,000 participants (95% CI: 8–33, $I^2$ = 54%). No biopsy-related deaths were reported.

### Conclusions

The proportion of participants with a valid follow-up biopsy in therapeutic trials in non-alcoholic steatohepatitis is on average 82%, with around 1 in 10 participants declining a follow-up biopsy. These findings can inform adequately-powered trials.

**Data Availability Statement:** All relevant data are within the paper and its Supporting Information files.

**Funding:** This study was funded by the National Institute for Health Research (NIHR) Oxford Biomedical Research Centre (grant number: IS-BRC-1215–20008). S.A.J. and P.A. are NIHR Senior Investigators and also funded by the NIHR Applied Research Collaboration Oxford and Thames Valley. EM is funded by a Wellcome Trust Clinical Doctoral Research Fellowship. The funder had no role in the design and conduct of the study; collection, management, analysis, and interpretation of the data; preparation, review, or approval of the manuscript; and decision to submit the manuscript for publication. The views expressed are those of the authors and not necessarily those of the NHS, the NIHR, or the Department of Health and Social Care.

**Competing interests:** The authors have declared that no competing interests exist.

# Introduction

Non-alcoholic steatohepatitis (NASH) is a chronic and progressive form of non-alcoholic fatty liver disease (NAFLD). It can progress to cirrhosis and end-stage liver disease, with resource implications for health care systems [1]. It is estimated to affect 6% of adults worldwide [2]. Patients with NASH have higher liver-related and cardiovascular morbidity and mortality compared with patients with early-stage NAFLD and the general population [3–5].

Many pharmacological agents are currently under development and testing in randomized clinical trials, but there is currently no medication licensed for the treatment of NASH. In these trials, assessment of changes in disease activity requires a liver biopsy at baseline and end of treatment, because there are no validated non-invasive biomarkers [6]. Based on the liver biopsy, regulators, such as the Food and Drug Administration and the European Medical Agency, currently consider specific histological changes in the liver after at least one year as acceptable intermediary endpoints in phase 2b-3 trials that are likely to predict progression to clinical outcomes [7, 8].

However, liver biopsies are invasive, and carry risks for patients, including pain, haemorrhage, and even death, and, thus, are unpopular with patients [9]. Achieving high follow-up is crucial to valid interpretation of data from clinical trials, so assessing the proportion of patients that undergo biopsy is imperative for trialists. Trials to date report variable follow-up and it is unclear what factors affect follow-up proportions or how best to estimate likely completion in future trials [10–13]. Moreover, it is crucial to understand whether patients are willing to undergo biopsy solely for the benefit of medical science, particularly for patients in control groups who have not received therapy. Some trials, such as those providing behavioural treatments such as weight loss, cannot be blinded and people in the control group could be deterred from a second biopsy by virtue of being offered no or inferior treatment. This review, therefore, aims to quantify the proportion of participants in NASH trials with a valid follow-up biopsy to inform future trial sample size estimates and assess why biopsy data is not obtained.

# Methods

This was a prospectively registered systematic review (PROSPERO ID: CRD42020189488) reported using the PRISMA guidelines. We followed the protocol with no changes except a minor revision described in the next section.

## Criteria

We included randomized clinical trials on adults with biopsy-confirmed non-alcoholic steatohepatitis with any degree of liver fibrosis, including no fibrosis and cirrhosis. The trials needed to evaluate any intervention and any comparator, ask participants to undertake a repeat liver biopsy within a minimum of one year from baseline (defined as at least 48 weeks), and report the number of participants analysed. The primary outcomes were the percentage of participants with a valid biopsy at follow-up and the percentage of participants declining a follow-up biopsy. We reworded the second primary outcome from 'attending a follow-up biopsy' in the protocol to 'declining a follow-up biopsy' in the paper to accurately reflect the way these data were reported in the primary publications. Secondary outcomes included reasons for lack of a valid follow-up biopsy, reimbursement for study participation, and biopsy-related complications.

## Search

We searched MEDLINE and Embase from inception until 26 May 2020 with no language restrictions. An experienced librarian developed the search strategy (available in the S1 File). We hand-searched studies from relevant systematic reviews.

## Extraction and assessment

Two researchers independently screened each title and abstract and each full text using an online standardized tool [14]. Following this, two researchers independently extracted the following data: baseline characteristics (age, sex, NAFLD activity score, fibrosis stage), nature of intervention, number of people with a biopsy at baseline and follow-up, reasons for lack of follow-up biopsy, biopsy-related adverse events, length and timings of follow-up, and reimbursement for study participation. These were pre-specified in the published protocol from the published papers and reported results in the clinical trial registries using a pre-defined and pre-piloted data extraction form and assessed risk of bias based on blinding of participants and personnel using the Cochrane Risk of Bias assessment [15]. We decided a priori not to assess risk of bias due to randomization and allocation, as we did not consider that they could meaningfully bias the primary outcome which itself is part of the tool (i.e. as attrition bias). Instead, we assessed risk of bias due to blinding of participants and personnel as high, low, or uncertain. Conflicts between reviewers were resolved through discussion or referral to a third reviewer.

## Analysis

We meta-analysed the data using generalized linear mixed models with random effects [16]. Statistical heterogeneity was assessed with the $I^2$ statistic. Prediction intervals to allow estimation of proportions in future studies were calculated [17]. Primary outcomes are summarized as proportions [95% confidence intervals (CI)]. Secondary outcomes are summarized as events per 1,000 participants (95% confidence intervals) given the low event probabilities. We excluded participants who had been randomized within a study that terminated early and therefore could not have a follow-up biopsy. If a study had conducted biopsies at two follow-up time points, we included the first time point in the primary analysis. We conceptualized the reasons for not undergoing a second biopsy in five categories: (a) as participant-related (i.e. consent withdrawal, explicitly declining biopsy, lost to follow-up, and moved away), (b) protocol-related (i.e. protocol deviation, did not start treatment, non-compliant), (c) medical-related (i.e. adverse events, physician decision, contraindications such as pregnancy, too ill, and death), and (d) trial-related [i.e. randomization by mistake, biopsy taken but not valid (e.g. due to inadequate tissue obtained), site closure], and (e) other/unclear (i.e. unclear, discharged from military). Publication bias was assessed with funnel plots [18].

Pre-specified subgroup analysis was conducted by location (high-income vs. middle-income countries) and time since first biopsy (1 year vs. more than 1 year). Pre-specified analysis was conducted comparing the proportion of valid follow-up biopsies between active intervention and placebo arms in placebo-controlled trials using odds ratios with 95% confidence intervals to examine evidence that blinding was implemented successfully and did not lead to differential dropout. For 3-arm and 4-arm trials, the data for the active arms were combined.

We also ran a post-hoc sensitivity analysis to assess whether proportions for follow-up biopsy differed between trial arms in open-label studies. This examined different potential expectation of benefit, where participants in a control or lower-intensity treatment arm might be less inclined to undergo a second biopsy due to perceived lack of expected benefit. For this analysis, we included open-label studies that compared (a) any medication plus diet and exercise advice against only diet and exercise advice, (b) intensive behavioural weight management programmes against basic diet and exercise advice, (c) two/three medication against one/two medication, and (d) a medication against a dietary supplement). We also ran a post-hoc meta-regression to examine if study size moderated retention and a subgroup analysis by geographical region. Analysis was conducted using the meta and metafor packages in RStudio v1.2.5033.

## Results

We screened 839 references and included 41 studies with a total of 6,695 participants (PRISMA Flowchart in Fig 1) [10–13, 19–55]. Seven studies were conducted in middle-income countries with the remaining run in high-income countries. Thirty-six studies tested medication as monotherapy or polytherapy with other medication or dietary supplements, four tested dietary supplements, and one tested a behavioural weight loss programme. Thirty-one trials had a placebo control arm, five trials had diet and exercise advice as control arms, and five trials compared two or more different drugs or supplements. Twenty-four trials were two-arm, 15 were three-arm, and two were four-arm. Among the 3-arm and 4-arm trials, twelve trials investigated various doses of the same medication whereas five trials compared multiple medications as monotherapy or polytherapy. S1 Table in S1 File presents the detailed study characteristics. At baseline, participants were on average 53.8 (SD: 11.4) years and 51% were female. They had mean NAFLD activity score and fibrosis stage of 5.11 (SD: 1.27) and 2.30 (SD: 1.15), respectively.

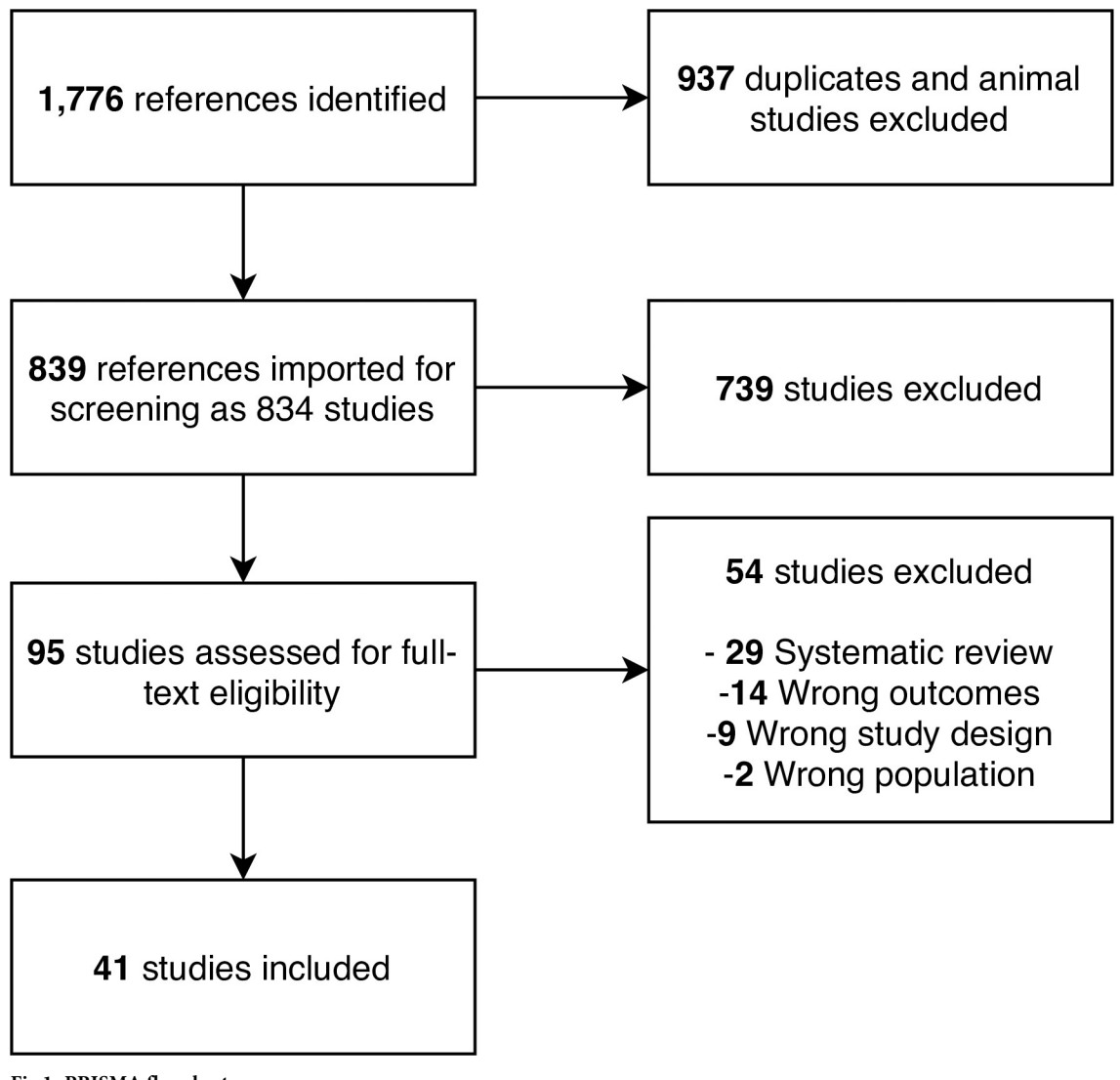

**Fig 1. PRISMA flowchart.**

Across all studies, the percentage of participants with a valid follow-up biopsy was 82% (95% CI: 78%–86%, $I^2$ = 92%, prediction interval: 50–96%) (Fig 2). Where it was reported, 9% (95% CI: 4–18%, $I^2$ = 92%, n = 13) of participants declined a biopsy (Fig 3). Among all studies, the main reasons for not having a valid follow-up biopsy were participant-related occurring in 95 per 1,000 participants (95% CI: 69–129, $I^2$ = 92%) (S1 Fig in S1 File).

Other reasons leading to a lack of a follow-up biopsy were medical-related [16 per 1,000 participants (95% CI: 8–29, $I^2$ = 91%)], protocol-related [3 per 1,000 participants (95% CI: 1–8, $I^2$ = 86%)], trial conduct-related [1 per 1,000 participants (95% CI: 0–5, $I^2$ = 91%)], and other/ unclear [3 per 1,000 participants (95% CI: 1–10, $I^2$ = 94%)] (S1-S5 Figs in S1 File).

There was no evidence that the location (high-income vs. middle-income counties) influenced the proportion of valid follow-up biopsies (S6 Fig in S1 File). In the analysis by geographical region, there was some limited evidence that this proportion was lower in two small studies conducted in Eastern Mediterranean but not in any other region (S7 Fig in S1 File). Moreover, there was no evidence that the interval to the follow-up biopsy (1 year vs. more than 1 year) affected this proportion (S8 Fig in S1 File). However, among the three studies that conducted an additional follow-up liver biopsy within 3 years, the proportion of participants with a second follow-up valid biopsy was 68% (95% CI: 61–75%, $I^2$ = 48%) when the proportion for the first follow-up biopsy was 89% (95% CI: 80–94%, $I^2$ = 74%) in these trials (S9 Fig in S1 File). Twenty-five trials reported the proportion of a follow-up biopsy by trial arm. Among these, there was no evidence of a difference in the proportion of valid follow-up biopsies between the active intervention and placebo arms (OR: 1.03, 95% CI: 0.87–1.22, $I^2$ = 0%) (S10 Fig in S1 File). Only one study with a high retention rate reported reimbursing participants for completing the study [55].

There were 32 studies that blinded personnel and participants and where it is reasonable to assume participants expected equal benefit between arms (risk of bias due to blinding in S2 Table in S1 File). There were eight open-label medication studies where expectations of benefit may or may not have been equal, and one open-label trial assessing a behavioural weight loss programme where blinding is by default not possible and where participants in the control arm presumably expected a second biopsy would not show large changes in histology. Among the open-label trials, there was no evidence that the proportion of valid follow-up biopsies differed between arms (OR: 1.11, 95% CI: 0.72–1.70, $I^2$ = 0%) (S11 Fig in S1 File). There was no evidence from a post-hoc meta-regression that the size of the study was associated with retention (b = 0.0009, 95% CI: -0.0001–0.0020, p = 0.08). The funnel plot showed no evidence of publication bias (S12 Fig in S1 File).

Biopsy-related serious adverse events occurred in 16 per 1,000 participants (95% CI: 8–33, $I^2$ = 54%) among the seven studies that specifically reported this outcome (Fig 4). Among the 17 serious adverse events, 4 were pain, 3 were hematoma, 1 was bile leak, and 9 were not reported. In 4 studies (n = 618) reporting a total of 27 non-serious adverse events, 25 were pain and 2 were vasovagal syncope. No biopsy-related deaths were reported.

## Discussion

This systematic review and meta-analysis of 41 randomized controlled trials in NASH shows that the proportion of participants with a valid follow-up biopsy is on average 82% (78–86%), but with unexplained heterogeneity between trials. On average, nearly one in 10 participants declined a second biopsy. About 3 out of 10 did not have a further follow-up biopsy. There was no evidence that the proportion having a follow-up biopsy differed by location, length of follow-up, between intervention and placebo groups in blinded trials, or between groups in trials with open allocation.

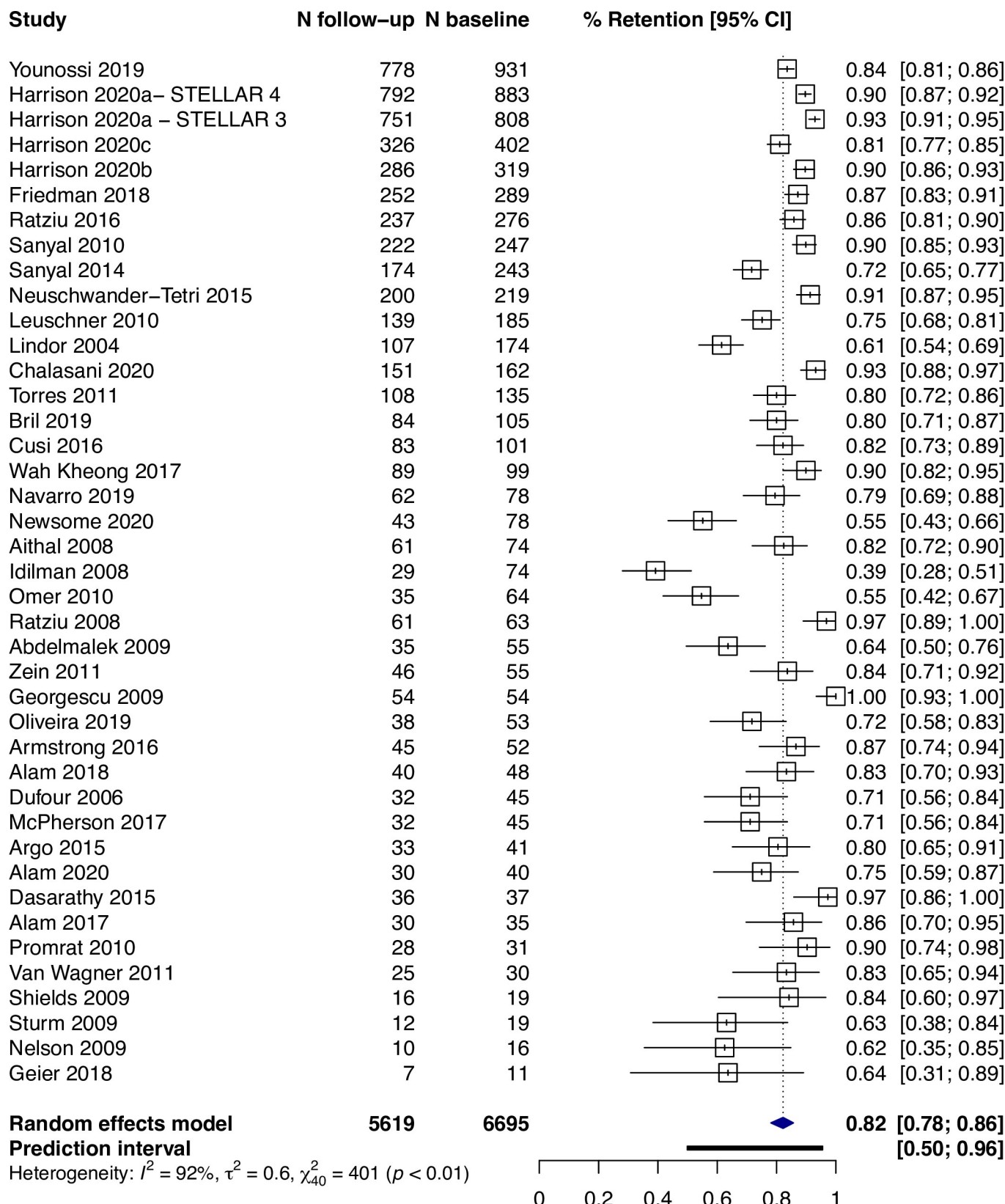

| Study | N follow-up | N baseline | % Retention [95% CI] |
|---|---|---|---|
| Younossi 2019 | 778 | 931 | 0.84 [0.81; 0.86] |
| Harrison 2020a– STELLAR 4 | 792 | 883 | 0.90 [0.87; 0.92] |
| Harrison 2020a – STELLAR 3 | 751 | 808 | 0.93 [0.91; 0.95] |
| Harrison 2020c | 326 | 402 | 0.81 [0.77; 0.85] |
| Harrison 2020b | 286 | 319 | 0.90 [0.86; 0.93] |
| Friedman 2018 | 252 | 289 | 0.87 [0.83; 0.91] |
| Ratziu 2016 | 237 | 276 | 0.86 [0.81; 0.90] |
| Sanyal 2010 | 222 | 247 | 0.90 [0.85; 0.93] |
| Sanyal 2014 | 174 | 243 | 0.72 [0.65; 0.77] |
| Neuschwander–Tetri 2015 | 200 | 219 | 0.91 [0.87; 0.95] |
| Leuschner 2010 | 139 | 185 | 0.75 [0.68; 0.81] |
| Lindor 2004 | 107 | 174 | 0.61 [0.54; 0.69] |
| Chalasani 2020 | 151 | 162 | 0.93 [0.88; 0.97] |
| Torres 2011 | 108 | 135 | 0.80 [0.72; 0.86] |
| Bril 2019 | 84 | 105 | 0.80 [0.71; 0.87] |
| Cusi 2016 | 83 | 101 | 0.82 [0.73; 0.89] |
| Wah Kheong 2017 | 89 | 99 | 0.90 [0.82; 0.95] |
| Navarro 2019 | 62 | 78 | 0.79 [0.69; 0.88] |
| Newsome 2020 | 43 | 78 | 0.55 [0.43; 0.66] |
| Aithal 2008 | 61 | 74 | 0.82 [0.72; 0.90] |
| Idilman 2008 | 29 | 74 | 0.39 [0.28; 0.51] |
| Omer 2010 | 35 | 64 | 0.55 [0.42; 0.67] |
| Ratziu 2008 | 61 | 63 | 0.97 [0.89; 1.00] |
| Abdelmalek 2009 | 35 | 55 | 0.64 [0.50; 0.76] |
| Zein 2011 | 46 | 55 | 0.84 [0.71; 0.92] |
| Georgescu 2009 | 54 | 54 | 1.00 [0.93; 1.00] |
| Oliveira 2019 | 38 | 53 | 0.72 [0.58; 0.83] |
| Armstrong 2016 | 45 | 52 | 0.87 [0.74; 0.94] |
| Alam 2018 | 40 | 48 | 0.83 [0.70; 0.93] |
| Dufour 2006 | 32 | 45 | 0.71 [0.56; 0.84] |
| McPherson 2017 | 32 | 45 | 0.71 [0.56; 0.84] |
| Argo 2015 | 33 | 41 | 0.80 [0.65; 0.91] |
| Alam 2020 | 30 | 40 | 0.75 [0.59; 0.87] |
| Dasarathy 2015 | 36 | 37 | 0.97 [0.86; 1.00] |
| Alam 2017 | 30 | 35 | 0.86 [0.70; 0.95] |
| Promrat 2010 | 28 | 31 | 0.90 [0.74; 0.98] |
| Van Wagner 2011 | 25 | 30 | 0.83 [0.65; 0.94] |
| Shields 2009 | 16 | 19 | 0.84 [0.60; 0.97] |
| Sturm 2009 | 12 | 19 | 0.63 [0.38; 0.84] |
| Nelson 2009 | 10 | 16 | 0.62 [0.35; 0.85] |
| Geier 2018 | 7 | 11 | 0.64 [0.31; 0.89] |
| **Random effects model** | **5619** | **6695** | **0.82 [0.78; 0.86]** |
| **Prediction interval** | | | **[0.50; 0.96]** |

Heterogeneity: $I^2 = 92\%$, $\tau^2 = 0.6$, $\chi^2_{40} = 401$ ($p < 0.01$)

**Fig 2. Proportion of participants (95% CI) with a valid follow-up biopsy.**

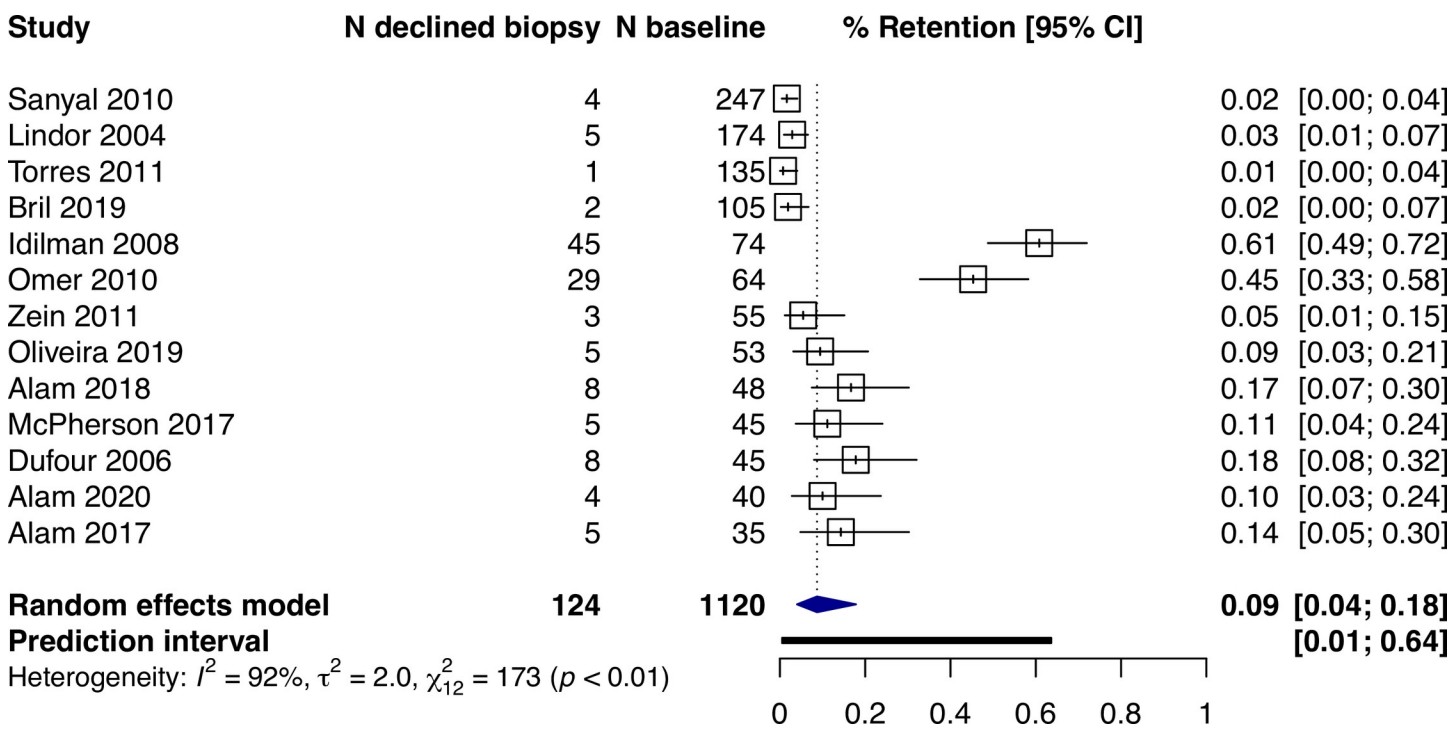

**Fig 3. Proportions of participants (95% CI) reporting to specifically decline a follow-up biopsy.**

These findings have implications for trial design of therapeutic trials in NASH, as the retention is essential for a sample size calculation. Firstly, many large-scale trials calculated their sample size based on a retention of 85–90%, but on average did not achieve this [11, 12, 39, 42, 47]. This can lead to underpowered studies that might not be able to answer their research questions, therefore, wasting resources and exposing participants to risky treatments and procedures. Future trials should allow for a lower retention to increase the likelihood of

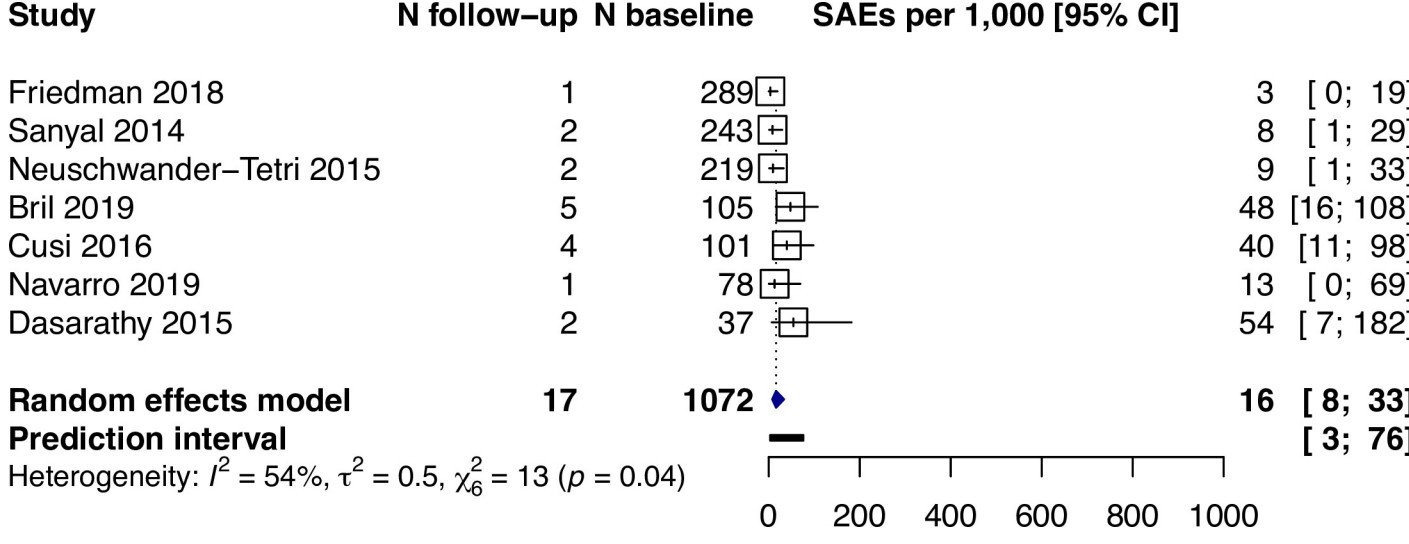

**Fig 4. Biopsy-related serious adverse events reported for N per 1,000 participants (95% CI).**

adequately powered studies. Secondly, the lack of evidence that the proportion of follow-up biopsy differ between arms in open label trials (i.e., where the risk of bias due to blinding was high) suggests that knowledge of the received treatment does not influence participants' commitment to the second biopsy. This might have been expected if participants perceived they had not received an effective treatment. Thirdly, the sample size should be based on expected histological changes in the first follow-up biopsy given the lower retention in subsequent follow-up biopsies.

Knowledge of the proportion of biopsy-related adverse events specifically in this population could be used as part of the consent process in future trials. Participants can be informed that about 1 in 60 participants may experience a serious adverse event, with the possibility of serious pain occurring in about 1 in 270 and of hematoma in about 1 in 350. These are generally in line with previous estimates from two previous audits totalling 3,482 biopsies in diverse populations [56, 57]. These audits reported either no deaths or 3 deaths in inpatients with major comorbidities. No study in this review reported a biopsy-related death. This is reassuring and might reflect the strict trial eligibility criteria that exclude people with major comorbidities. Non-invasive biomarkers, such as magnetic resonance imaging and elastography, are emerging as alternatives to the liver biopsy that will minimise such adverse events [58, 59]. The new proposed definition of metabolic dysfunction-associated fatty liver disease [60] might reinforce the use of such biomarkers for diagnosis and as trial endpoints [58], although the optimal terminology remains to be determined [61].

Strengths of the study include being guided by Cochrane methods and including randomized trials of diverse interventions with a minimum follow-up of 1 year. The estimates remained robust irrespective of time to the first follow-up or location indicating that they should be applicable to studies worldwide. However, the study also has some limitations. Most studies did not report on biopsy-related adverse events and, thus, it remains unclear if participants in these studies did not experience these or if studies only reported on treatment-related adverse events. Our estimates include only studies specifically reporting adverse events and, thus, assume that the decision to report biopsy-related events is unrelated to their occurrence in these trials. Future studies should present detailed analyses of all adverse events.

## Conclusions

The proportion of participants with a valid follow-up biopsy in therapeutic trials in NASH is on average 82%, with around 1 in 10 participants declining a follow-up biopsy. This provides a realistic assessment of likely follow-up to inform sample size calculations in future trials.

## Supporting information

**S1 Checklist. PRISMA 2009 checklist.**
(DOC)

**S1 File.**
(PDF)

## Acknowledgments

We would like to thank Nia Roberts, the librarian who constructed and ran the search.

## Author Contributions

**Conceptualization:** Dimitrios A. Koutoukidis.

**Formal analysis:** Dimitrios A. Koutoukidis.

**Funding acquisition:** Susan A. Jebb, Paul Aveyard.

**Investigation:** Dimitrios A. Koutoukidis, Elizabeth Morris, John A. Henry, Yusra Shammoon, Matthew Zimmerman, Moscho Michalopoulou.

**Methodology:** Dimitrios A. Koutoukidis, Susan A. Jebb, Paul Aveyard.

**Supervision:** Susan A. Jebb, Paul Aveyard.

**Writing – original draft:** Dimitrios A. Koutoukidis.

**Writing – review & editing:** Dimitrios A. Koutoukidis, Elizabeth Morris, John A. Henry, Yusra Shammoon, Matthew Zimmerman, Moscho Michalopoulou, Susan A. Jebb, Paul Aveyard.

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
