## [Decision Letter · Decision Letter 0]

8 Mar 2021

PONE-D-21-01926

What proportion of people have a follow-up biopsy in randomized trials of treatments for non-alcoholic steatohepatitis?: A systematic review and meta-analysis

PLOS ONE

Dear Dr. Koutoukidis,

Thank you for submitting your manuscript to PLOS ONE. After careful consideration, we feel that it has merit but does not fully meet PLOS ONE’s publication criteria as it currently stands. Therefore, we invite you to submit a revised version of the manuscript that addresses the points raised during the review process.

We look forward to receiving your revised manuscript.

Kind regards,

Jee-Fu Huang, M.D., Ph.D.

Academic Editor

PLOS ONE

2. Please described the data extraction methods in more details. We would expect to see reporting of the specific information extracted from the manuscripts.

Reviewers' comments:

Reviewer's Responses to Questions

5. Review Comments to the Author

Reviewer #1: In this paper, Koutoukidis et al. assessed quantify the proportion of participants with a valid follow-up biopsy. They found that the proportion of participants with a valid follow-up biopsy was 82%. There was no evidence of a difference by location, trial length, or by allocated treatment group. Reasons for missing follow-up biopsies were, in ranked order, related to participants, medical factors, protocol, trial conduct, and other/unclear. The present study deals with an important topic and may provide important information for the clinical trial for NAFLD/NASH. This meta-analysis will be improved by revising following issues.

Major comments

#1. As the results showed, repeated biopsy could not be performed in approximately 20% of enrolled subjects even in clinical trials. These findings indicated the importance of non-invasive tests as a main outcome. Recently, magnetic resonance elastography (MRE) is reported have high diagnostic accuracy (PMID: 29908362, PMID: 31556124, PMID: 30291868, PMID: 30014566). The authors should discuss the importance of non-invasive tests in clinical trials by referring the above reports.

#2. Recently, a new definition for metabolic dysfunction-associated fatty liver disease (MAFLD) has been proposed by an international expert consensus panel (PMID: 32278004). In MAFLD criteria, liver biopsy is not required for the diagnosis and MAFLD is associated with hepatic fibrosis than NAFLD (PMID: 32478487, PMID: 32997882) indicating that non-invasive tests/metabolic dysfunction can be main outcome in the MAFLD definition. The authors should discuss the future perspective in MAFLD ear.

#3. Meta-regression analysis is better to be added in this study.

Minor comments

#1. In conclusion (line 58), the authors conclude that “…is on average 82%, with around 1 in 10 participants declining..”. It should be “approximately 2 in 10 participants declining..”

#2. Quality assessment should be performed according to the criteria formulated by the Cochran Effective Practice and Organization of Care (EPOC) group. The authors should assess the followings: random sequence generation, allocation concealment, blinding of participants and researchers, blinding of outcome assessment, incomplete outcome data, selective reporting, other bias.

Reviewer #2: This is a systematic review and meta-analysis regarding what proportion of participants have a follow-up biopsy in randomized trials of treatments for non-alcoholic steatohepatitis (NASH). There are forty-one randomized clinical trials in NASH with a total of 6,695 participants included in this systematic review. The results demonstrated that the proportion of participants with a valid follow-up biopsy was on average 82% (78-86%), but with unexplained heterogeneity between trials. The reasons for missing follow-up biopsies were related to participants, medical factors, protocol, trial conduct, and other/unclear reasons. Biopsy-related serious adverse events occurred in 16 per 1,000 participants.

Liver biopsies are invasive and have potential risks including pain, bleeding, bile leak and even death for patients. Participants particularly in placebo control groups who did not receive therapy may hesitate to undertake a follow-up liver biopsy.

It is important to know that whether participants are willing to undergo a follow-up liver biopsy are also affected by geographic and cultural differences. However, these factors are not take into consideration in this review.

---

## [Author Response · Author response to Decision Letter 0]

25 Mar 2021

Please see the relevant uploaded document for our point-by-point response.

---

## [Decision Letter · Decision Letter 1]

6 Apr 2021

What proportion of people have a follow-up biopsy in randomized trials of treatments for non-alcoholic steatohepatitis?: A systematic review and meta-analysis

PONE-D-21-01926R1

Dear Dr. Koutoukidis,

We’re pleased to inform you that your manuscript has been judged scientifically suitable for publication and will be formally accepted for publication once it meets all outstanding technical requirements.

Kind regards,

Jee-Fu Huang, M.D., Ph.D.

Academic Editor

PLOS ONE

Additional Editor Comments (optional):

Reviewers' comments:

Reviewer's Responses to Questions

**Comments to the Author**

1. If the authors have adequately addressed your comments raised in a previous round of review and you feel that this manuscript is now acceptable for publication, you may indicate that here to bypass the “Comments to the Author” section, enter your conflict of interest statement in the “Confidential to Editor” section, and submit your "Accept" recommendation.

Reviewer #1: All comments have been addressed

---

## [Editor Report · Acceptance letter]

12 Apr 2021

PONE-D-21-01926R1 

What proportion of people have a follow-up biopsy in randomized trials of treatments for non-alcoholic steatohepatitis?: A systematic review and meta-analysis 

Dear Dr. Koutoukidis:

I'm pleased to inform you that your manuscript has been deemed suitable for publication in PLOS ONE. Congratulations! Your manuscript is now with our production department. 

Kind regards, 

on behalf of

Dr. Jee-Fu Huang 

Academic Editor

PLOS ONE